# Determinant Factors of Post-Partum Contraception among Women during COVID-19 in West Java Province, Indonesia

**DOI:** 10.3390/ijerph20032303

**Published:** 2023-01-28

**Authors:** Laili Rahayuwati, Ikeu Nurhidayah, Rindang Ekawati, Habsyah Saparidah Agustina, Dadang Suhenda, Dean Rosmawati, Vira Amelia

**Affiliations:** 1Department of Community Nursing, Faculty of Nursing, Universitas Padjadjaran, Bandung 45363, Indonesia; 2Department of Pediatric Nursing, Faculty of Nursing, Universitas Padjadjaran, Bandung 45363, Indonesia; 3National Population and Family Planning Board, Jakarta 13650, Indonesia; 4Department of Mental Health Nursing, Faculty of Nursing, Universitas Padjadjaran, Bandung 45363, Indonesia; 5Faculty of Nursing, Universitas Padjadjaran, Bandung 45363, Indonesia

**Keywords:** age of first marriage, personal support, post-partum contraception

## Abstract

Background: One of the manifestations of family development is pregnancy planning, where this method is applied 0–42 days after childbirth. Post-partum contraception is an effort to avoid pregnancy by using contraceptive medicine from 42 days to 84 days after childbirth. Purpose: This research aims to analyze the attitudes of fertile couples who use contraceptive devices after childbirth during the COVID-19 pandemic and the factors that influence it. Method: This research uses a quantitative method approach. The sampling technique was random sampling with proportional sampling so that 280 respondents were obtained from 3 regencies/cities in West Java with high fertility rates and low post-partum contraceptive participation rates. Quantitative data analysis used univariate, bivariate, and multivariate methods. Result: The results showed that the final model of the analysis of the most determining factors for post-partum contraception during a pandemic were family support, healthcare staff support, counselling with healthcare staff, attitudes, and age at first marriage. Conclusion: Fertile couples with the highest amount of family support are more likely to use post-partum contraceptive devices during COVID-19. The results of this study can be used as material for consideration in making decisions about post-partum contraception, especially during the COVID-19 pandemic.

## 1. Introduction

The National Population and Family Planning Board predicted an increase in unwanted pregnancies in Indonesia and an increase of contraceptives in the COVID-19 era [1]. The family planning program in Indonesia serves 28 million people, but about 10% who received the benefits in 34 provinces found it difficult to access the program in March and women found it more difficult to receive information and services they needed because of the concerning situation of the health system and staff worldwide [2].

According to the National Population Board, the decrease of contraceptive uses for one month in Indonesia can add a risk of pregnancy to 15%, resulting in about 420,000 pregnancies in one to three months. If people keep losing access to contraceptives, the pregnancy rate can rise up to 30%, resulting in 800,000 pregnancies in the months to come, while many health centers are closed and those that are open are limiting the number of patients [1,3].

One of the manifestations of family development is pregnancy planning. One of the government’s programs is family planning, which is an effort to avoid pregnancy by using contraceptive medicine starting from 42 days to 84 days after childbirth. Meanwhile, Post-Miscarriage Family Planning is what prevents mothers from having a miscarriage for up to 14 days. The use of contraception is vital to avoid any unwanted pregnancies with a close interval from previous pregnancies, which is one of the 4T components (too young, too old, too many, and too close) [4].

Such a risky situation of pregnancy can lead to many complications, which eventually can contribute to the death of mother and child. Therefore, post-partum contraception is one of the more strategic efforts to reduce infant mortality, maternal mortality, and the total fertility rate (TFR). The aim of the program is to avoid any unwanted pregnancies, and to adjust the spacing of one pregnancy to another, so that families can plan for a safe and healthy pregnancy [5].

The choice of contraceptive is closely related to the husband’s support or partner’s approval [6]. The husband’s role in family planning can be realized directly or indirectly. Direct participation can be realized by becoming a family planning acceptor while husband’s indirect participation is to support the wife in family planning, namely, as a motivator and decision-maker, in order to plan the number of children in the family. The role of the husband as a motivator is by providing encouragement to become a contraceptive participant [6]. The support that is given by the husband to strengthen the use of contraception to his wife includes accompanying her during counselling, utilizing contraceptives, accompanying controls, and always protecting her when something undesirable happens [7].

The participation of fertile couples remains a priority for population control. According to 2017 data, the contraceptive prevalence rate in Indonesia was 64% and there was a 57% decrease in modern contraceptives in women who got married at the age of 15–49 years old [8].

The participation of fertile couples can be influenced by various factors, which can be reviewed through the precede-proceed attitude theory [9]. This consists of predisposing factors, enabling factors, and reinforcing factors. Predisposing factors are those which come from within the individual, including knowledge, behavior, beliefs, and values [10].

Other studies show that enabling factors come from outside the individual. The fact that the contraceptive program is not affordable and a lack of the program services for men is related to the low level of participation of men as contraceptive acceptors [11,12].

The third factor is reinforcing factors, such as family support, which has an impact on the participation of couples in using contraceptives. Another reinforcing factor is support from local public figures, both formal figures such as institutions or informal ones such as religious figures, the youth community, or the elderly [13].

It is explained that the West Java province is the most populated province in Indonesia. Therefore, this research was conducted in three regencies/cities with the highest fertility rate and lowest participation rate toward post-partum contraception. In addition, the explanation on predisposing factors, enabling factor, and reinforcing factor, needs to be identified to control the birth rate amidst the COVID-19 pandemic. Therefore, it is vital to analyze the predictor factors that influence participation in contraceptive use. This is to be a point of consideration for population and family planning programs.

## 2. Methods

### 2.1. Study Design and Setting

This research used a quantitative method with a cross-sectional approach.

### 2.2. Sample Size and Sampling Technique

The total population in this research was 9476. The sampling technique was done using random sampling with proportional cluster sampling so that 280 respondents were obtained from three regencies/cities in West Java with high fertility rates and low post-partum contraceptive participation rates. To determine the number of samples from the total population with a significance level of 10%, we used the Isaac and Michael table; the minimum sample size was 263.

### 2.3. Inclusion Criteria and Exclusion Criteria

The inclusive criteria were women aged 15–49 years old, have given birth in the last eight months or during COVID-19, and with good access to technology. The number of quantitative samples was around 100 per regency/city. Out of the expected 300 respondents, only 280 were selected as the quantitative samples. The exclusion criteria were women who have labor complications and are unable to use contraception devices.

### 2.4. Data Collection Tools

Quantitative data collection was carried out using self-administered questionnaires that were developed and validated by the National Population Board experts. Quantitative data collection was carried out using paper questionnaires and online questionnaires. The respondents were guided by data collectors on how to fill out the questionnaires. 

Quantitative data analysis was done using univariate analysis; bivariate analysis to identify the impact between individual factors, predisposing factors, socio-demographic, and external factors; enabling factors and reinforcing factors toward couple participation in post-partum contraception; and multivariate analysis to identify the most dominant factors toward the post-partum contraception. 

### 2.5. Ethical Considerations

This research has received approval from The Ethical Committee of Universitas Padjadjaran, with number: 939/UN6/KEP/EC/2020. This research was conducted from May to December 2020.

## 3. Results

The results are displayed in several tables. Table 1 describes the frequency distribution of the respondents.

Table 1 shows the results of the analysis of frequency distribution of marital age; the information contraceptive during pregnancy, during labor, and after labor; contraceptive counselling; reason for not taking the program; type of program; place for taking contraceptive; and recommendation. The analysis results showed that almost half of the respondents (49.6%) got married at the age of 17–21 years old, the information about the program was obtained during pregnancy (85%) and during labor (87.5%). The source of information was received from the nurse (34.3) and more than one source (40.7%). 

Table 2 shows the results of the distribution for predisposing factors, enabling factors, and reinforcing factors on family planning program. The analysis results of the predisposing factors shows that more than half of the respondents had good knowledge (55%) and had an attitude that was aligned with the knowledge (59.3%). Meanwhile, the analysis results of enabling factors showed that most respondents (68.2%) were afforded a facility, access, and healthcare staff. Most respondents (57.5%) claimed that the availability of facilities in their regions was adequate.

The analysis results of reinforcing factors shows that nearly all respondents (95%) who had the post-partum program obtained family support, whether it was from the husband, parents, or other family members. In addition, nearly half of the respondents (29.3%) claimed that the information source was from midwives.

Table 3 shows the results of the analysis of the relationship between predisposing factors, enabling factors, and reinforcing factors using the program. From the table, it is identified that the demographic factors had a significant relationship with the age of first marriage (*p*-value = 0.03). Meanwhile, other demographic factors such as income (*p*-value = 0.477), education (*p*-value = 0.119), and children (*p*-value = 0.725) did not show any significant relationship with the use of post-partum contraceptives.

Other factors such as contraceptive information during pregnancy (*p*-value = 0.042), contraceptive counselling during pregnancy (*p*-value = 0.002), contraceptive information after labor (*p*-value = 0.023), contraceptive information source after labor (*p*-value = 0.037), and contraceptive counselling to healthcare staff (*p*-value = 0.000) had a significant relationship with the use of post-partum contraceptives.

In Table 4, the final modelling of the analysis of the determinants of the post-partum programs during the pandemic is the largest OR value, that is those supported by family had a 11.589 chance of using contraception. Meanwhile, the support from healthcare staff had an OR = 2.910, counselling with healthcare staff was OR = 2.493, and age of first marriage was OR = 0.599.

## 4. Discussion

Based on the data analysis on this research, there are four most dominant factors that determine a person using the post-partum program, namely family support, healthcare staff support, counselling, couple’s attitude, and the first age of marriage.

Family support has a *p*-value of 0.023, which means that it is impactful to the use of the post-partum program. Research that was conducted by Tran et al. [14] shows that family support, especially from the husband, to the mothers after childbirth is the most impactful factor. In undergoing family planning, husband support is vital. In Indonesia, it has been shown that it is the husband who gives permission to the wife whether or not to use contraception or to take a post-partum program. 

Other research shows that there were cases of 20 women who used contraceptives influenced by their partners. A common feature of the women was education level; all but one of them had less than high school education. A total of four women could not choose bilateral tubal ligation (BTL) contraception because of the domination of their husbands. There were 16 women that reported that their husbands did not want to use condoms as a method of contraception, so these women chose one of the other methods. The research that was conducted by Kahramanoglu et al. shows that the permission or support from the husband has a vital impact for the mothers. Other research also shows support from the husband to use post-partum contraception because the husband wants to avoid pregnancy [15].

The second most impactful factor toward the use of post-partum contraception is healthcare staff with the *p*-value 0.001, which means that it is impactful to the use of post-partum contraception [14,16]. Research that was conducted by Kahramanoglu et al. [15] showed that almost all participants (99.7%) in the study agreed that healthcare staff were the best source for information about contraception. The effect of counselling depends on what information is given and how. Client-emphasized protective care is proposed to enhance the care experience and women’s ability to achieve their own reproductive goals. The necessary steps are defined as providing friend-like interactions with women, listening to women to find out what is most important to them about family planning methods, and providing relevant information according to their preferences [17].

The next most impactful factor on the use of post-partum is counselling. Based on the data analysis, the *p*-value for counselling was 0.006. This shows that there is an impact of counselling toward the participation of post-partum contraception. This research result is supported by Abbas et al. [18], saying that counselling that was provided for post-partum mothers influences their participation in post-partum contraception significantly [19,20,21,22]. Contraceptive counselling has great potential as a strategy to empower women who do not want pregnancy to choose a method of family planning that she can use correctly and consistently over time, thereby reducing the individual’s risk of unwanted pregnancy [23].

The fourth most dominant factor toward the use of post-partum contraception is the first age of marriage. The research results showed that 139 respondents got married for the first time at the age of 17 to 21 years old, and that age is the most dominating of all. The *p*-value in this research was 0.004 for first age of marriage. This means that it was the most impactful factor towards the use of post-partum contraception [24,25,26,27]. This shows that there is an impact of age on the use of post-partum contraception. This is in line with the research from Nansseu et al. [28]. 

Research by Mowafi and Elde [29] found that the use of contraceptive method is lower among early marriages before 18 years compared to those who were married after 18 years of age. Age has a positive relationship with the choice of contraceptive method, where a high level of maturity of the reproductive system or the age of the mother will be followed by an increase in the choice of long-term contraceptive methods [30].

In the era of COVID-19, the Population Board predicted the increase in unwanted pregnancies and the decrease in contraceptives among fertile couples. Family support, especially husband support, becomes the key factor of the involvement in post-partum contraception. Counselling and education on programs should be targeted not only to fertile women, but also to their partners, so that they can give support to participate in post-partum contraception. In addition, healthcare staff support is also an important reinforcing factor that affects the mother’s attitude toward post-partum contraception. Therefore, positive and persuasive motivation from them can be a turning point for those mothers. Moreover, the research results can be a reference of consideration in making decisions regarding post-partum contraception, especially during the COVID-19 pandemic.

The study has limitations, including: (1) the process of listing respondents through an online system and (2) inter-sectoral collaboration, in terms of providing information on maternal data copy of the last six months, has an impact on data collection.

## 5. Conclusions

The increase in unwanted pregnancies and the decrease in contraception in couples was predicted during COVID-19. Therefore, it is vital to plan the pregnancy and determine how many children are in the family; one choice is post-partum contraception. Family support is the most dominant factor related to the participation of couples in post-partum contraception. The support from the husband is the key factor in whether a couple wants to use post-partum contraception or not. The second factor is healthcare staff, coming from nurses, doctors, family program staff, and nurses in post-partum rooms. 

Family support, especially the husband, becomes the key factor in the involvement of couples in post-partum contraception. Counselling and education on family programs should be targeted not only to women but also their partners. Additionally, healthcare staff support is also an important reinforcing factor that affects the women’s attitude toward post-partum contraception.

## Figures and Tables

**Table 1 ijerph-20-02303-t001:** Frequency distribution of marriage age, information during pregnancy, information during labor, information source, counselling, reason for not taking contraception, types of contraception, and places (n = 280).

Variables	Frequency	Percentage (%)
Using Contraception:		
No	89	31.8
Yes	191	68.2
Age of First Marriage		
<17 years old	18	6.4
17–21 years old	139	49.6
22–26 years old	98	35.0
27–31 years old	20	7.1
>32 years old	5	1.8
Contraceptive Information During Pregnancy:		
Received	42	15.0
Did Not Receive	238	85.0
Contraceptive Information After Labor:		
Received	30	10.7
Did Not Receive	250	89.3
Contraceptive Information Source During Pregnancy:		
Did Not Receive Any	27	9.6
Midwife	96	34.3
Clinic Staff	8	2.9
General Practitioner/Gynecologist	12	4.3
Hospital/Clinic	6	2.1
Electronic/Non-Electronic Media	3	1.1
Friends/Neighbors/Parents/Cadre	14	5.0
Other Sources	114	40.7
Contraceptive Counselling:		
No	62	22.1
Yes	218	77.9
Reason for Not Taking Contraception:		
Contraceptive User	191	68.2
Want to Have Another Child Soon	1	0.4
Too Old	4	1.4
Rarely Hang Out	6	2.1
Know Its Side Effects	24	8.6
Health Reasons	15	5.4
Husband Using Contraception	4	1.4
Myth/Taboo	2	0.7
More Than One Reason	33	11.8
Type of Contraception:		
No Contraception	89	31.8
Condom	2	0.7
IUD/Spiral	32	11.4
Pill	7	2.5
Injection	145	51.8
Implant	5	1.8
Places for Taking Contraception		
No Contraception	89	30.7
Hospital	13	4.6
General Clinic	39	13.9
Community Health Center	5	1.8
Contraceptive Clinic/Midwife Practitioner	132	47.1
Pharmacy Shop	2	0.7
Contraceptive Recommendation:		
No Contraception	89	30.4
Own Initiative	139	49.6
Husband	27	9.6
Parents/Parents-In-Law/Other Family Members	5	0.17
Healthcare Staff (Doctor, nurse, midwife)	20	7.1

**Table 2 ijerph-20-02303-t002:** Distribution of predisposing factors, enabling factors, and reinforcing factors on post-partum contraception (n = 280).

Variables	Frequency	Percentage (%)
Predisposing Factor		
Knowledge:		
Fair	126	45.0
Good	154	55.0
Attitude:		
Positive	114	40.7
Negative	166	59.3
Enabling Factor		
Facility, Access, and Healthcare Staff Support		
None	89	31.8
Fulfilled	191	68.2
Contraceptive Facilities		
Inadequate	119	42.5
Adequate	161	57.5
Distance to Contraceptive Service Facility		
Far	105	37.5
Medium-Near	175	62.5
Access to Contraceptive Service Facility		
Difficult	9	3.2
Easy	271	96.8
Reinforcing Factor		
Family Support		
None	14	5.0
Fulfilled	266	95.0
Healthcare Staff Support		
None	80	28.6
Fulfilled	200	71.4
Getting Information on Contraceptive During COVID-19		
None	66	23.6
Fulfilled	214	76.4
Information Source on Contraceptive During COVID-19		
None	66	23.6
Midwife	82	29.3
Clinic staff/Contraceptive staff	10	3.6
General Practitioner/Gynecologist	7	3.5
Hospital/Clinic	5	1.8
Electronic/Non-Electronic Media	4	1.4
Friends/Neighbors/Families	15	5.4
More Than One Source of Information	91	32.5

**Table 3 ijerph-20-02303-t003:** The relationship between predisposing factors, enabling factors, and reinforcing factors using the program.

Variables	Contraceptive Use				*p*-Value
	Yes		No		
	Frequency	Percentage (%)	Frequency	Percentage (%)	
Age of First Marriage					
<17 years old	14	77.8	4	22.2	
17–21 years old	105	75.5	34	24.5	
22–26 years old	57	58.2	41	41.8	0.03
27–31 years old	11	55.0	9	45.0	
32 years old	4	80.0	1	20.0	
Income					
<1,000,000	17	77.3	5	22.7	
1,000,001–2,000,000	85	70.2	36	29.8	
2,000,001–3,000,000	62	68.9	28	31.1	0.477
3,000,001–4,000,000	13	65.0	7	35.0	
4,000,001–5,000,000	11	52.4	10	47.6	
>5,000,001	3	50.0	3	50.0	
Educational					
Primary School	14	77.8	4	22.2	
Elementary School	70	73.5	23	24.7	
High School	90	63.8	51	36.2	
Diploma	8	80.0	2	20.0	0.119
Bachelor	8	47.1	9	52.9	
Magister/Doctoral	1	100.0	0	0.0	
Children					
1–2	140	67.6	67	32.4	0.725
>2	51	69.9	22	30.1	
Contraceptive Information During Pregnancy					
Received	168	70.6	70	29.4	0.042
Did Not Receive	23	54.8	19	45.2	
Counselling During Pregnancy Check					
Yes	175	71.4	70	28.6	0.002
No	16	45.7	19	54.3	
Contraceptive Information After Labor					
Received	176	70.4	74	29.6	0.023
Did Not Receive	15	50.0	15	50.0	
Contraceptive Information Source After Labor					
Did Not Receive Any	13	48.1	14	51.9	
Midwife	71	75.0	24	25.0	
Clinic Staff	7	87.5	1	12.5	
General Practitioner/Gynecologist	5	41.7	7	58.3	
Hospital/Clinic	4	66.7	2	33.3	0.037
Electronic/Non-Electronic Media	2	66.7	1	33.3	
Friends/Neighbors/Parents/Cadre	7	50.0	7	50.0	
Other Sources	81	71.1	33	28.9	
Contraceptive Counselling To Healthcare Staff					
Yes	161	68.2	89	31.8	0.000
No	30	48.4	32	51.6	

**Table 4 ijerph-20-02303-t004:** Modelling of determinant factors on choosing post-partum contraceptives during the COVID-19 Pandemic.

Variables	B	SE	Wald	OR (95% CI)	*p*-Value
Family Support	2.450	1.078	5.168	11.589	0.023
Healthcare Staff Support	1.068	0.311	11.788	2.910	0.001
Counselling with Healthcare Staff	0.913	0.331	7.629	2.493	0.006
Age of First Marriage	−0.512	0.177	8.369	0.599	0.004

## Data Availability

The data presented in this study are available on request from the corresponding author. The data are not publicly available due to privacy from the respondents and data is confidential according to ethics.

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
