# Peer review of "Determinant Factors of Post-Partum Contraception among Women during COVID-19 in West Java Province, Indonesia"

_ijerph, 2023, doi:10.3390/ijerph20032303_

Round 1
Reviewer 1 Report
The authors presented the problem od post-partum contraception in Indonesia, which is a specific problem of the region. The manuscript covers an important problem for the countries with high fertility rates only. It requires extensive English language corrections.
Several parts of the article are not clear to me and require explanations:
1. „ by using contraceptive medicine from 42 days to 6 weeks after childbirth” – I do not understand – 42 days is exactly 6 weeks…
2. Table 1: Contraceptive Information During Pregnancy: it is repeated twice in the table, however different results are given – please explain (probably should be: during labour)
3. Table 2: what do you mean by “fair” and “good” knowledge? What were the measures of this factor? Knowledge of what exactly?
4. For the paragraph” From the table, it is identified that the demographic factors had a significant relationship with the age of first 131 marriage (p-value = 0,03). Meanwhile, other demographic factors such as income (p-value - 0.477), education (p-value = 0.119) and children (p-value = 0.725) did not show any significant relationship with the use of post-partum contraceptives. Other factors such as contraceptive information during pregnancy (p-value = 0.042), contraceptive counselling during pregnancy check (p-value = 0.003), contraceptive information after pregnancy (p-value = 0.022), contraceptive information source after labor (p-value = 0.037), and contraceptive counselling to healthcare staff (p-value = 0,000) had a significant relationship with the use of post-partum contraceptives.”- there are no numbers presented and it is difficult to verify what data gave the results of significant/non-significant p-values. They should be presented in form of a table instead of listing percentages in them only.
5. Please explain: The total population in this research was 9476 respondents…..Out of the expected 300 respondents, only 280 were selected as the quantitative samples. – Why was the difference so great between 9476 respondents and the analysis on only 280? I do not understand.
6. “Other research shows that the permission or support from the husband has a vital impact for the mothers. Other research also shows support from husband to use post-partum contraception because the husband wants to dis-158 miss the pregnancy.[13]” – the authors mention “other researches” but they are not discussing the numbers with their own work – the discussion is therefore incomplete and improper. The same comment refers to all following paragraphs of the discussion.
7. The information about cultural differences in Indonesia should be added to the introduction (husbands’ decision and allowance for the contraception) – it is not obvious for people from other countries, especially in Europe or US.
8. The conclusions should be shortened as they are the repetition of results and parts of discussion.
Author Response
Thanks for all the suggestions and input from the reviewers. We appreciate and use it for the improvement of our manuscript. All changes in our manuscript are marked in yellow.
Response to Reviewer 1 Comments
Poin 1:„ by using contraceptive medicine from 42 days to 6 weeks after childbirth” – I do not understand – 42 days is exactly 6 weeks…
Response 1: Thank you for pointing it out. Based on your inquiry, we would like to revise the period from 42 days to 84 days. In other words, it is calculated starting from 42 days after giving birth to 84 days later, still in the post-partum period.
Point 2: Table 1: Contraceptive Information During Pregnancy: it is repeated twice in the table, however different results are given – please explain (probably should be: during labour)
Response 2: Thank you for this advice. A typing error has occurred. As for what we meant about contraceptive information in the table is, it should be a contraceptive during pregnancy and after labour. We have revised the table.
Point 3:Table 2: what do you mean by “fair” and “good” knowledge? What were the measures of this factor? Knowledge of what exactly?
Response 3: Thank you for the question. We divided knowledge into fair and good. Questions about knowledge related to pregnancy and contraception. It includes several indicators regarding pregnancy plans, how many pregnancies, efforts made during pregnancy in maintaining health in terms of nutrition, activities, and understanding of post-pregnancy, including the use of contraception, and how to maintain post-pregnancy health for both mother and child
Point 4:For the paragraph” From the table, it is identified that the demographic factors had a significant relationship with the age of first 131 marriage (p-value = 0,03). Meanwhile, other demographic factors such as income (p-value - 0.477), education (p-value = 0.119) and children (p-value = 0.725) did not show any significant relationship with the use of post-partum contraceptives. Other factors such as contraceptive information during pregnancy (p-value = 0.042), contraceptive counselling during pregnancy check (p-value = 0.003), contraceptive information after pregnancy (p-value = 0.022), contraceptive information source after labor (p-value = 0.037), and contraceptive counselling to healthcare staff (p-value = 0,000) had a significant relationship with the use of post-partum contraceptives.”- there are no numbers presented and it is difficult to verify what data gave the results of significant/non-significant p-values. They should be presented in form of a table instead of listing percentages in them only.
Response 4: Thank you for the review that has been done. Some of the revisions we did are as follows: We have presented the related figures according to the reviewer's suggestions so that the p-values ​​are significant/insignificant. In table 3, several changes in related variables are relevant to the existing explanations. In addition, after we reviewed the statistical results, there was a change in the P-Value for contraceptive counseling during pregnancy check (p-value = 0.002), contraceptive information after labor (p-value = 0.023)
Point 5: Please explain: The total population in this research was 9476 respondents…..Out of the expected 300 respondents, only 280 were selected as the quantitative samples. – Why was the difference so great between 9476 respondents and the analysis on only 280? I do not understand.
Response 5: We appreciate the suggestion. As it should be, 9476 participants are the total population and not respondents. The sampling technique was done using random sampling in order to obtain 280 respondents from 3 districts/cities in West Java with high fertility rates and low post-partum family planning participation rates. Determining the number of samples from the total population with a significance level of 10% using the Isaac and Michael table, the minimum sample size is 263.
Point 6: “Other research shows that the permission or support from the husband has a vital impact for the mothers. Other research also shows support from husband to use post-partum contraception because the husband wants to dis-158 miss the pregnancy.[13]” – the authors mention “other researches” but they are not discussing the numbers with their own work – the discussion is therefore incomplete and improper. The same comment refers to all following paragraphs of the discussion.
Response 6: Indeed, did some additional information and references, among others are:
Other research shows that there were cases of 20 women who used contraceptives influenced by their partners. A common feature of the women was education level: all but one of them had less than high school education. Four women could not choose bilateral tubal ligation (BTL) contraception because of the domination of their husbands. Sixteen women reported that their husbands did not want to use condoms as a method of contraception, so these women chose one of the other methods [1].
Contraceptive counselling has great potential as a strategy to empower women who do not want pregnancy to choose a method of family planning that she can use correctly and consistently over time, thereby reducing the individual's risk of unwanted pregnancy [2]
Research by Mowafi and Elde [29] found that the use of contraceptive method is lower among early marriages before 18 years compared to those who were married after 18 years of age. Age has a positive relationship with the choice of contraceptive method, where a high level of maturity of the reproductive system or the age of the mother will be followed by an increase in the choice of long-term contraceptive methods [3]
References:
- Kahramanoglu I, Baktiroglu M, Turan H, Kahramanoglu O, Verit FF, Yucel O (2017) What influences women’s contraceptive choice? A cross-sectional study from Turkey. Ginekol Pol 88:639–646
- Dehlendorf C, Krajewski C, Borrero S (2014) Contraceptive counseling: Best practices to ensure quality communication and enable effective contraceptive use. Clin Obstet Gynecol 57:659–673
- Yusni P, Yusni I (2021) The Effect of Counseling on Contraceptive Selection in Women of Reproductive Age Couples. Sci Midwifery 9:252–259
Point 7: The information about cultural differences in Indonesia should be added to the introduction (husbands’ decision and allowance for the contraception) – it is not obvious for people from other countries, especially in Europe or US.
Response 7: Certainly. We added several studies related to culture, specifically about the situation in Indonesia related to the role of husband and wife in the family.
Point 8: The conclusions should be shortened as they are the repetition of results and parts of discussion.
Response 8: Thank you. We try to summarize the conclusions as follows:
The increase in unwanted pregnancies and decrease in contraception in couples was predicted during COVID-19. Therefore, it is vital to plan the pregnancy and determine how many children are in the family; one choice is post-partum contraception. Family support is the most dominant factor related to the participation of couples in post-partum contraception. The support from husband is the key factor in whether a couple wants to use post-partum contraception or not. The second factor is healthcare staff, coming from nurses, doctors, family program staff, and nurses in post-partum rooms.
Family support, especially the husband, becomes the key factor of the involvement of couples in post-partum contraception. Counselling and education on family programs should be targeted not only to women but also their partners. Additionally, healthcare staff support is also an important reinforcing factor that affects the women’s attitude toward post-partum contraception

Reviewer 2 Report
Title of the manuscript should be more explanatory (it should include the place where the study done and the target population)
Validation of the used questionnaire is not present
How you calculate sample size?
Where is the statistical tests used in the present study?
Materials and methods should be divided in section: study design and setting- sample size calculation- sampling technique- inclusion criteria- exclusion criteria- data collection tool- statistical analysis and ethical considerations
Knowledge and attitude in table (2): knowledge of what?
How you assess this knowledge and attitude?
Table (4): need statistical consultation (where are the reference groups of the determinant factors?)
Discussion is very short and weak
Author Response
Thanks for all the suggestions and input from the reviewers. We appreciate and use it for the improvement of our manuscript. All changes in our manuscript are marked in yellow.
Response to Reviewer 2 Comments
Point 1: Title of the manuscript should be more explanatory (it should include the place where the study done and the target population)
Response 1:
Thank you for the title suggestion. We added according to suggestions. We’ve changed the title with a specific place and target population.
The Tittle:
Determinant Factors of Post-Partum Contraception among Women during COVID-19 in West Java Province, Indonesia
Point 2: Validation of the used questionnaire is not present
Response 2: Thank you for the observation.
Validation on the use of the instrument has been carried out by the National Population Board. The instrument trial with 47 questions was declared valid using a comparison of r count compared to r table, the result was 0.227 with a significance degree of 0.05. Some invalid question items are not used.
Point 3: How you calculate sample size?
Response 3: Thank you for the question.
With regard to the number of samples, referring to the sampling reference compared to the existing population, the sampling technique was done using random sampling with proportional cluster sampling so that 280 respondents were obtained from 3 regencies/cities in West Java with high fertility rates and low post-partum contraceptive participation rates. Determining the number of samples from the total population with a significant level of 10% used the Isaac and Michael table, a minimum sample size was 263.
Point 4: Where is the statistical tests used in the present study?
Response 4: In previous studies,
Chandler's research on contraceptive use involved 172 female veterans in this project.
While research from Smitha, a descriptive study was conducted among 110 married women attending Obstetrics and Gynecology.
As for the Gotwal study, A cross-sectional descriptive study of 173 nursing staff using a structured questionnaire on knowledge, attitude scale, and practice and preference were done
Point 5: Materials and methods should be divided in section: study design and setting- sample size calculation- sampling technique- inclusion criteria- exclusion criteria- data collection tool- statistical analysis and ethical considerations
Response 5: Thank you. We revise the parts according to suggestions. We have changed and rewritten the “material and method” according to the section.
Study Design and Setting
This research used a quantitative method with a cross-sectional approach.
Sample Size and Sampling Technique
The total population in this research was 9476. The sampling technique was done using random sampling with proportional cluster sampling so that 280 respondents were obtained from three regencies/cities in West Java with high fertility rates and low post-partum contraceptive participation rates. Determining the number of samples from the total population with a significance level of 10% used the Isaac and Michael table, a minimum sample size was 263.
Inclusion Criteria and Exclusion Criteria
The inclusive criteria were women aged 15 - 49 years old, have given birth in the last eight months or during COVID-19, and have good access to technology. The number of quantitative samples was around 100 per regency/city. Out of the expected 300 respondents, only 280 were selected as the quantitative samples. Exclusion criteria were women who have labor complications and are unable to use contraception device.
Data Collection Tools
Quantitative data collection was carried out using self-administered questionnaires developed and validated by the National Population Board experts. Quantitative data collection was carried out using paper questionnaires and online questionnaires. Respondents were guided by data collectors on how to fill out the questionnaires.
Quantitative data analysis was done using univariate analysis; bivariate analysis to identify the impact between individual factors, predisposing factors, socio-demographic, and external factors; enabling factors and reinforcing factors toward couple participation in post-partum contraception, and; multivariate analysis to identify the most dominant factors toward the post-partum contraception.
Ethical Considerations
This research has received approval from The Ethical Committee of Universitas Padjadjaran, with number: 939/UN6/KEP/EC/2020. This research was conducted from May to December 2020.
Point 6: Knowledge and attitude in table (2): knowledge of what?
Response 6: An explanation of the knowledge we use is:
Knowledge of pregnancy
Post-partum contraception
Attitude toward postpartum contraception
Point 7: How you assess this knowledge and attitude?
Response 7: Using knowledge and attitude instruments about contraceptive use that have been validated by the National Population Board
Point 8: Table (4): need statistical consultation (where are the reference groups of the determinant factors?)
Response 8: The statistical results obtained show that the determinant factor in this study was a factor that has a significant relationship with contraceptive use
Point 9: Discussion is very short and weak
Response 9: We reviewed the discussion. The additions are as follows:
Other research shows that there were cases of 20 women who used contraceptives influenced by their partners. A common feature of the women was education level: all but one of them had less than high school education. Four women could not choose bilateral tubal ligation (BTL) contraception because of the domination of their husbands. Sixteen women reported that their husbands did not want to use condoms as a method of contraception, so these women chose one of the other methods [1].
Contraceptive counselling has great potential as a strategy to empower women who do not want pregnancy to choose a method of family planning that she can use correctly and consistently over time, thereby reducing the individual's risk of unwanted pregnancy [2]
Research by Mowafi and Elde [29] found that the use of contraceptive method is lower among early marriages before 18 years compared to those who were married after 18 years of age. Age has a positive relationship with the choice of contraceptive method, where a high level of maturity of the reproductive system or the age of the mother will be followed by an increase in the choice of long-term contraceptive methods [3]
References:
- Kahramanoglu I, Baktiroglu M, Turan H, Kahramanoglu O, Verit FF, Yucel O (2017) What influences women’s contraceptive choice? A cross-sectional study from Turkey. Ginekol Pol 88:639–646
- Dehlendorf C, Krajewski C, Borrero S (2014) Contraceptive counseling: Best practices to ensure quality communication and enable effective contraceptive use. Clin Obstet Gynecol 57:659–673
- Yusni P, Yusni I (2021) The Effect of Counseling on Contraceptive Selection in Women of Reproductive Age Couples. Sci Midwifery 9:252–259

Round 2
Reviewer 1 Report
Dear authors,
You have changed and improved your manuscript according to my review. Table 3 is now much better and gives clear information. It was necessary to clarify cultural differences in the introduction, as well as to explain the population recruitment for the study - it was done. The discussion was improved - you are still missing something in line 188 (..other research conducted by (.[15] showed...) - I think you missed the name of the authors - it needs corrections. The additional literature is quite new and relevant.
I only believe the paper requires some English language editing.
I have no further comments.
Author Response
Dear Reviewers,
Thank you for giving us the opportunity to resubmit and revised our manuscript. From your suggestion, we made corrections to the comments previously made. In the manuscript, we have highlighted the second revised manuscript in green. We hope that our revisions and responses to this manuscript will be accepted for further consideration and publication
Response to Reviewer 1
Point 1: You have changed and improved your manuscript according to my review. Table 3 is now much better and gives clear information. It was necessary to clarify cultural differences in the introduction, as well as to explain the population recruitment for the study - it was done. The discussion was improved - you are still missing something in line 188 (..other research conducted by (.[15] showed...) - I think you missed the name of the authors - it needs corrections. The additional literature is quite new and relevant
Response 1:Thank you for carefully reviewing our manuscript. It was our mistake, we missed to write the author's name. We already revised the statement bellow:
Research conducted by Kahramanoglu et al. [15] showed that almost all participants (99.7%) in the study agreed that healthcare staff were the best source for information about contraception.
Point 2: I only believe the paper requires some English language editing.
Response 2: Based on reviewers' comments regarding the English language, we have improved the English language in the first round of revision through professional proofreading.
Reviewer 2 Report
The manuscript is now clear in all aspects
Author Response
Thank you very much for your meticulous review and valuable suggestions.